# The Development of a New Tool to Help Patients and Their Providers Evaluate Self-Management of Type 2 Diabetes Mellitus

**DOI:** 10.3390/healthcare11152117

**Published:** 2023-07-25

**Authors:** Asma Obad, Ravneet Singh, Saara Nasruddin, Erin Holmes, Meagen Rosenthal

**Affiliations:** Department of Pharmacy Administration, School of Pharmacy, University of Mississippi, University, MS 38677, USA

**Keywords:** Type 2 Diabetes Mellitus, diabetes self-management, diabetes self-care

## Abstract

Diabetes self-management or self-care activity related to diet, physical activity, and glucose monitoring, among other things, is recognized as important to effectively managing this condition. The aim of this study was to create an assessment tool for evaluating knowledge and self-management behavior in Type 2 Diabetes Mellitus (T2DM) for patients and their providers. The study utilized an online survey with a cross-sectional design of adults diagnosed with Type 2 diabetes. The survey consisted of 8 sections and a total of 56 questions, which were designed to measure the participants’ current knowledge and behavior regarding diabetes self-management. The total sample size was 306 participants, and the results revealed a significant association between performance on diabetes knowledge questions and self-management behavior (β = 0.46; 95% CI: 0.34, 0.58; *p* < 0.001). Furthermore, education had a significant impact on diabetes self-management behavior (β = 0.59; 95% CI: 0.14, 1.03; *p* = 0.01). Overall, the data indicated that participants who performed well on knowledge-based questions exhibited higher scores in desired diabetes management behaviors. Increasing awareness of this work in the diabetic community could facilitate the clinical encounters between diabetic patients and their healthcare providers, with an emphasis on each individual’s needs being taken into consideration.

## 1. Introduction

Diabetes Mellitus (DM) is caused by irregular blood glucose levels due to insulin deficiencies. DM is the seventh leading cause of death in the United States, and patients diagnosed with diabetes have a 50% higher risk of early death compared to those without [1]. Previous data report an increased number of patients with diabetes of roughly 54% since 2015 along with an estimated 38% increase in diabetes-related deaths annually [2]. According to the Centers for Disease Control and Prevention (CDC), approximately 88 million adults were classified as having pre-diabetes in 2018, which translates to 1 in every 10 people in the United States being diagnosed with some form of diabetes [3]. 

According to the CDC, 34.2 million people have been diagnosed with Type 2 diabetes [3]. To decrease the number of deaths or serious complications in patients with Type 2 diabetes (T2DM), it is important that those diagnosed are aware of the disease and how to engage in the various self-care activities needed to effectively manage the condition [1]. These self-care activities include, but are not limited to, eating properly, exercising, practicing good hygiene and foot care, and continuously monitoring blood glucose levels [4]. Diabetes self-management education and support (DSMES) tools have been suggested as effective ways to increase patients’ knowledge of their diagnosis. The DSMES are tools or services aimed at preventing or delaying complications due to diabetes. DSMES training must meet the standards of the American Diabetes Association (ADA) and the American Association of Diabetes Educators (AADE) in order to “ensure that services are offering quality education and make the services eligible for reimbursement from Medicare, many private health plans, and some state Medicaid agencies [1]”. However, each patient manages the disease differently depending on various factors, such as health literacy, family support, emotional state, financial support, level of income, and where they live. Bains et al. surveyed 125 adults from a primary care clinic and found that 68.2% of participants in the study had less than a high school education, 64.2% had a household income of <$15,000, and 73.9% that their health status for the year that the research was performed was worse than the previous year [5]. It is essential to note that a high percentage of these patients with lower incomes are likely to have more pressing concerns, such as earning a living, looking for a place to live, and earning an income for their families’ survival [5]. Therefore, effective yet personalized approaches based on these external factors are necessary to make sure that the patient has a comprehensive understanding of T2DM.

There are many programs that have developed diabetes self-knowledge tools to evaluate a patient’s understanding of their disease and how to self-manage it. The Diabetes Self-Management Questionnaire (DSMQ) and the Diabetes Knowledge Questionnaire (DKQ) have been tested and demonstrated reliability [6,7]. The DSMQ is composed of 16 questions and focuses on participants’ perceptions of diabetes self-management behavior [6]. The DKQ consists of two sections and focuses on assessing patient knowledge of diabetes [7]. While each of these measures offers valuable insights into patient knowledge and behavior, they are not intended to be used by clinicians during patient encounters to guide patient-tailored educational or behavioral interventions. 

A similar critique might also be offered of current DSMES training programs. The programs we mentioned in the previous paragraph do result in significant improvements, but they have low access and retention rates for clinical measures such as HbA1c. Furthermore, these programs consist of classes that cover very broad topic areas and follow strict schedules that may not be convenient for all the participants [8,9]. These programs are also designed to inform patients about particular pre-chosen topics that the clinicians and developers deem important regarding diabetes self-management, but that information may not be the information that the patient wants or even lacks [8,9] This restrictive schedule and curriculum limit patients’ choices on topics that they would like to further emphasize or the order in which they would like to receive the information. To our knowledge, there has yet to be a measure that allows patients to evaluate their own knowledge and behavior deficits around diabetes self-management and then allows them to choose the order in which they would like to cover the various self-management topics that are unique to their individual needs. Therefore, the objective of this study is to develop a diabetes self-management assessment tool that allows patients and their provers to evaluate T2DM knowledge and self-management behavior. 

## 2. Materials and Methods

### 2.1. Participants and Data

This study was conducted using a cross-sectional online survey of patients 18 years or older with T2DM in the United States. The patients were recruited via a panel from Amazon Mechanical Turk, or MTurk, a crowdsourcing website that allows “workers” to perform various tasks for a small commission [10]. The study was reviewed and exempted by the University of Mississippi Institutional Review Board (IRB) before survey administration (Protocol number: 21x-184). Inclusion criteria included United States adult residents who were 18 years or older with a diagnosis of T2DM. 

The minimum target sample size for this study was 150 participants per the sample size calculator G*Power for a linear multiple regression with six predictors, medium effect size, and power of 0.95 [11].

### 2.2. Survey Instrument

This survey contains 8 sections and a total of 56 questions designed to assess participants’ current knowledge and behavior toward diabetes self-management. These questions are derived from a previous study conducted at the University of Mississippi and include demographic questions about such subjects as gender, education, and urban/rural setting [12]. Each section is based on a core component of diabetes self-management skills [13]. In the first section, items are used to determine the participants’ prior knowledge of the disease process and treatment. The second section of the survey contains items assessing knowledge of healthy eating habits. The third section contains items about engagement in physical activities. In the fourth section, the participants answer items about monitoring blood sugars and using patient-generated health data in self-management decision-making. The fifth section contains items about the participants’ medication use. The sixth section asks participants how they would respond if certain diabetes complications arose. The seventh section evaluates how the participants would prevent, detect, and treat acute and chronic complications regarding their diabetes. Finally, the eighth section contains questions regarding healthy coping mechanisms with psychosocial issues and concerns. For these eight sections, there are two categories of questions. One category covers knowledge-based information that assesses whether participants know specific information related to their health such as “stress increases blood glucose levels”. The other category covers behavioral questions meant to evaluate if participants are performing behaviors that can positively affect their health such as “I track my glucose levels”.

### 2.3. Data Collection

The survey was posted as a “job” for completion on MTurk (Seattle, WA, USA). For this study, potential participants were screened for their age and self-reported diabetes status. After the screening portion on MTurk, the participants were directed to a Qualtrics survey link containing the complete survey. The first data import was on 5 April 2021 and the last data import was on 7 May 2021, once a sufficient sample size was achieved. There was a 20% fee on the reward, and a bonus amount (if any) when workers were paid. On MTurk, a question that needs to be answered is referred to as a Human Intelligence Task, or HIT [10]. HITs with 10 or more assignments were charged an additional 20% fee on the reward. Workers are paid. The minimum fee is $0.01 per assignment or bonus payment [10]. 

### 2.4. Data Analysis

All analysis procedures were performed on IBM SPSS v28.0.1.0. Knowledge and behavior questions were scored based on correct responses, and mean percent knowledge and behavior scores were calculated and reported. Means for respondents’ latest HbA1c tests and length of time since diagnosis with T2DM were calculated. Percentages of respondents’ gender identification, educational attainment, and geographic setting (urban/rural) were also reported. 

The performance of the knowledge and behavior questions was evaluated through an item analysis. Point biserial correlations were calculated for each question by correlating performance on the question to performance on the overall knowledge/behavior score. Point biserial correlations for each item were then compared to the total percentage of respondents who answered the question correctly using the DCOM Suggested Guidelines for Reviewing and Eliminating Question Items [14]. Items with a point biserial of 0.15 or greater indicated that respondents who had overall high behavior/knowledge scores answered those items correctly. Items were kept if the point biserial was 0.15 or greater and 50.1–100% of respondents correctly answered the question. According to the DCOM Suggested Guidelines for Reviewing and Eliminating Question Items, items outside of this range should be reviewed [14]. A point biserial ≥ 0.3 is high and indicates good discrimination between those who score “high” on the behavior/knowledge questions and those who score low on these questions. 

Questions that did not pass the item analysis were removed. The Kuder-Richardson Formula 20 (KR-20) was used to evaluate the reliability of the eight subscales. The relationship between knowledge scores on behavior scores was compared to assure an expected positive and significant correlation. Linear regression was used to predict diabetes self-management behavior scores based on knowledge scores performance while adjusting for the latest HbA1c results, length of time since diagnosis of T2DM, gender identification, educational attainment, and geographic setting.

## 3. Results

There was a total of 420 responses received through MTurk. However, after completing a list-wise deletion of missing cases, incomplete responses, and those who did not meet screening criteria, the total sample size was 306 participants. The sample population were mostly male (56.2%), had a master’s degree (55.9%), lived in an urban setting (78.4%), had an average HbA1c of 6.9, and were diagnosed with T2DM for approximately three years on average (Table 1). The average diabetes self-management knowledge score and behavior scores were 71.7% and 68.2%, respectively.

The item analysis for both knowledge and behavior scores shows that all questions included are statistically significant. Approximately 54 out of 56 items were retained in the analysis. One question was removed due to it not meeting the biserial criterion. Another question was removed due to incorrect coding between the instrument and the answer key. Two behavioral items, question 50 and question 54, had point biserial correlations within range but had low numbers of correct responses: 45.6% and 47.2%, respectively. After review, both items were retained in the analysis (Table 2 and Table 3). Table 4 provides reliability results (KR-20) for each of the 8 subscales as well as all behavioral items, knowledge items, and all 54 items. Each of the subscales exhibited poor reliabilities individually, with improvements shown when analyzing all behavioral or knowledge items as scales, respectively.

Performance on the diabetes knowledge questions was statistically significantly associated with diabetes self-management behavior questions (*β* = 0.46; 95% CI: 0.34, 0.58; *p* < 0.001), as shown in Table 5. A unit increase in diabetes knowledge score was associated with a 0.46 unit increase in diabetes self-management behavior score adjusting for the latest HbA1c results, length of time diagnosed with T2DM, gender, education, and geographic setting. Additionally, education significantly impacted diabetes self-management behavior (*β* = 0.59; 95% CI: 0.14, 1.03; *p* = 0.01). A unit increase in educational attainment was associated with a 0.59 unit increase in diabetes self-management behavior score adjusting for diabetes knowledge score, the latest HbA1c results, length of time diagnosed with T2DM, gender, and geographic setting. 

## 4. Discussion

The main objective of this project was to develop a tool to evaluate patients’ educational knowledge and behavior towards diabetes self-management for themselves and their providers. The ultimate goal of this work is to facilitate the clinical encounters between patients with diabetes and a healthcare provider, with an emphasis on the individual needs of the patient. It was found that participants that scored better on knowledge-based questions overall had greater scores in desired diabetes management behaviors. However, not all knowledge questions yielded the same increase in desired behaviors (Table 2). Lastly, it was found that an increase in knowledge scores and educational attainment was significantly correlated with an increase in diabetes management behaviors, whereas HbA1c scores, duration of diagnosis, gender, and geographical settings were not. It should be noted that 136 of our 306 participants did not recall or report their last HbA1c. This is somewhat consistent with other reports of patient recall rates of HbA1c [15]. Interestingly, findings by Willaing et al. also suggest that poor recall of HbA1c is related to poor diabetes self-management behaviors [15].

As stated in previous studies, diabetes knowledge is one of the most important factors associated with glycemic control, which was also supported in this study. Aids such as the one used in this study are useful in evaluating a patient’s previous understanding of T2DM and allow healthcare providers and patients to see and address knowledge gaps [6]. Previous instruments such as the DSMQ and DKQ were not designed to be used in clinical encounters with patients.

Given the performance of the instrument in this study, the next step will be to assess its performance in a clinical setting and allow patients to rank which of the knowledge deficit areas is of greatest interest to them and see how a clinician such as a community pharmacist may provide the needed education within their existing workflow. This gives patients more autonomy in learning how to better take care of themselves and helps healthcare providers structure a specialized care plan for them. Previous reviews and randomized control trials have shown how the interventions of pharmacists result in a significant effect of improvement in T2DM, its complications, and even long-term improvement [16,17,18]. A systematic literature review and a meta-analysis found that pharmacist-led self-management interventions improved the HbA1c (long-term clinical parameter for long-term diabetes follow-up) values in diabetic patients, which emphasizes the positive impact of pharmacists on the patient-healthcare provider relationship [16]. In a randomized controlled trial in patients with T2DM to look for improvement of diabetes self-management through a clinical pharmacy program, the authors found that patients who received an individualized pharmacotherapeutic care plan and diabetes education improved their medical knowledge, medication adherence, correct insulin injections, and monitoring blood glucose levels techniques compared to the control group. Moreover, the mean HbA1c values in this group decreased significantly compared to the control group [19]. Both of these studies concluded that the quality of life for these patients was significantly improved. An annual or biannual survey may be distributed to help patients assess areas that they may need more information on. According to a report from North Carolina, Medicaid patients have been shown to visit a community pharmacy significantly more than visiting a primary healthcare provider [20]. Thus, the pharmacy setting can allow patients to obtain vital information with less hassle and, therefore, pharmaceutical intervention can have a significant impact.

There may be some limitations to the study’s sample. MTurk participants may be less diverse than the general US population, resulting in a sample that is not representative of the population in general. Furthermore, recruiting could have been hindered because MTurk may have been unable to recruit people on the basis of characteristics that they have not profiled. As an example, recruiting underrepresented groups and diverse socioeconomic backgrounds may have been more difficult than recruiting other groups. This was evident in the very highly educated participants that completed our survey. Future research studies that administer this instrument should include participants of various education levels and backgrounds.

## 5. Conclusions

The primary aim of this research project was to develop a patient decision aid to facilitate the assessment of patients’ knowledge and behavior regarding diabetes self-management in order to improve patient outcomes. As a whole, our data showed that those participants who were able to perform better on knowledge-based questions had higher scores on the behaviors desired to manage diabetes. Increasing awareness of this work in the diabetic community could facilitate the clinical encounters between diabetic patients and their healthcare providers, with an emphasis on each individual’s needs being taken into consideration. 

## Figures and Tables

**Table 1 healthcare-11-02117-t001:** Descriptive Statistics.

	%	N
Gender		
Male	56.2	172
Female	40.8	125
Non-binary	2.6	8
Education		
High School	7.8	24
Bachelor’s Degree	9.8	30
Master’s Degree	55.9	171
Doctorate Degree	22.9	70
Other	2.9	9
Setting		
Urban	78.4	240
Rural	21.6	66
Knowledge % Score (mean)	71.7	306
Behavior % Score (mean)	68.2	306
Latest HbA1c test (mean) *	6.9	169
How long [in months] were diagnosed with T2DM (mean)	36.4	306

* 137 participants did not report their latest HbA1c.

**Table 2 healthcare-11-02117-t002:** Performance of Behavior Items.

		Total % Agreement	Point Biserial	*p*-Value
Participant agreement with the statement:	Subscale			
I do a good job managing my diabetes and use regular medication regimens, etc.	Diabetes disease process and treatment	53.4	0.257	<0.001
I understand what is happening to my body when I am experiencing high or low glucose levels.	Diabetes disease process and treatment	96.7	0.177	0.002
When interpreting nutritional information on a food label, I use the recommended servings of food on my meals based on the content on the label.	Healthy eating	84	0.366	<0.001
When choosing a protein for dinner, the majority of the time I eat chicken, turkey, and fish.	Healthy eating	60.3	0.43	<0.001
I exercise (definition: physical activity that follows a plan or schedule) generally twice a week or more.	Physical activity	64.5	0.429	<0.001
If exercising, I check my blood glucose levels.	Physical activity	54.1	0.498	<0.001
I currently have a food diary or app that helps keep track of my daily eating habits.	Monitoring and using patient-generated health data	54.4	0.209	<0.001
I have experience with viewing online test results.	Monitoring and using patient-generated health data	62.9	0.221	<0.001
I track my glucose levels.	Monitoring and using patient-generated health data	82.7	0.297	<0.001
I am confident in my ability to test my own glucose levels using a glucose meter.	Monitoring and using patient-generated health data	90.9	0.266	<0.001
I keep track of my medications by creating a routine, writing out my prescriptions, and storing medications in a pillbox.	Medication use	65.8	0.487	<0.001
When prescribed a medication, I usually read the side effects and the instructions that come with the medication.	Medication use	71.7	0.426	<0.001
I take my daily medication(s) regularly, as directed by my doctor/pharmacist.	Medication use	78.5	0.473	<0.001
When encountered with a problem managing your diabetes, I try my best to solve it and usually learn from it.	Problem solving	52.4	0.449	<0.001
If I develop the flu and notice that my blood glucose levels are higher than normal, I will monitor my diabetes more frequently, contact my healthcare provider, and research how the flu can affect my blood glucose levels.	Problem solving	56.7	0.406	<0.001
If I am on vacation at a hotel and do not have regular access to the gym, I will ask the front desk staff about local walking trails and try to walk as much as possible.	Problem solving	52.4	0.544	<0.001
I frequently communicate with a diabetes educator or healthcare professional.	Preventing, detecting, and treating acute and chronic complications	79.8	0.363	<0.001
I receive a flu shot every year.	Preventing, detecting, and treating acute and chronic complications	75.6	0.157	0.006
I do not currently smoke or have ever smoked.	Preventing, detecting, and treating acute and chronic complications	45.6	0.459	<0.001
When assessing the emotional impact of diabetes in my life, I usually have positive feelings.	Healthy coping with psychosocial issues and concerns	47.2	0.261	<0.001
My chosen coping mechanisms to deal with the effect of stress on my mind and body are relaxation, rest, spending time with family while maintaining a positive attitude towards life.	Healthy coping with psychosocial issues and concerns	76.9	0.426	<0.001
My attitude towards my diabetes treatment tends to be headstrong and committed to keeping up with my healthcare plan.	Healthy coping with psychosocial issues and concerns	53.1	0.403	<0.001
When diagnosed with diabetes, I reacted positively and optimistically.	Healthy coping with psychosocial issues and concerns	53.4	0.405	<0.001
When I think about the complications of the impact of diabetes on my life, I generally become eager to seek support from health professionals, family, and friends.	Healthy coping with psychosocial issues and concerns	69.1	0.292	<0.001

Legend: Sample size of 306; items included if point biserial ≥ 0.15 and percentage of respondents answering the question correctly ≥ 50.1%; items outside of these criteria were reviewed and/or eliminated.

**Table 3 healthcare-11-02117-t003:** Performance of knowledge items.

		Total % Agreement	Point Biserial	*p*-Value
Participant agreement with the statement:	Subscale			
High blood glucose is bad for my body.	Diabetes disease process and treatment	56	0.594	<0.001
Diabetes can affect nerves, kidneys, and the cardiovascular system.	Diabetes disease process and treatment	59.9	0.572	<0.001
Hemoglobin A1C test measures my average level of glucose in the blood in the past three months.	Diabetes disease process and treatment	57.7	0.386	<0.001
The following contain “carbs”: bread, pasta, fruits, dairy products, and sugary foods such as desserts.	Healthy eating	77.9	0.376	<0.001
Olive oil, canola oil, and fish oil are considered healthy.	Healthy eating	81.8	0.266	<0.001
According to a nutritional label, if a serving provides more than 20% of the recommended daily value, that food item is high in that nutrient.	Healthy eating	79.8	0.272	<0.001
The recommended method for cooking meat is roasting, boiling, and grilling.	Healthy eating	81.8	0.428	<0.001
When choosing a healthier meal plan, it is often recommended to increase the variety of the foods you eat.	Healthy eating	62.5	0.156	0.006
I should drink milk with skim or 1% fat content.	Healthy eating	77.9	0.324	<0.001
Most days of the week, half of my dinner plate should be filled with non-starchy vegetables such as fresh greens or broccoli.	Healthy eating	78.8	0.441	<0.001
Exercise does ALL of the following for the body: reduces blood glucose levels and the amount of insulin needed to control those levels, reduces pain and leg cramps, and improves moods.	Physical activity	70	0.427	<0.001
Physical activity is known as any movement that results in burning calories (such as walking upstairs, gardening, or performing housework). I usually participate in physical activity at least two days per week.	Physical activity	87	0.324	<0.001
Participating in physical activity usually lowers blood glucose levels.	Physical activity	84	0.348	<0.001
Self-monitoring diabetes is important because it allows you to facilitate your glucose changes with your lifestyle.	Monitoring and using patient-generated health data	52.8	0.455	<0.001
The two most important tests that you can accomplish at home to successfully manage your diabetes are a glucose meter and a blood pressure test.	Monitoring and using patient-generated health data	77.2	0.241	<0.001
A food diary is important because it helps you keep track of what you are eating and when your meals are.	Monitoring and using patient generated health data	53.7	0.586	<0.001
We take medications for diabetes to control blood glucose levels.	Medication use	81.8	0.438	<0.001
Diabetes medications perform all of the following functions: help the pancreas produce more insulin, help muscles become more sensitive to insulin, and limit the liver’s release of stored sugar.	Medication use	86.6	0.244	<0.001
Insulin is presented to the body by injection.	Medication use	65.8	0.475	<0.001
A dose of insulin is given according to your blood glucose levels.	Medication use	66.8	0.565	<0.001
Insulin needs to be injected just below the skin for slower absorption into the fat.	Medication use	55.4	0.438	<0.001
Taking too much diabetes medication or engaging in physical activity can cause you to experience low glucose problems.	Problem solving	74.3	0.236	<0.001
Hypertension is the term for blood pressure greater than or equal to 140/90 mm Hg.	Preventing, detecting, and treating acute and chronic complications	83.7	0.377	<0.001
Low doses of Aspirin can prevent heart attacks.	Preventing, detecting, and treating acute and chronic complications	85.3	0.347	<0.001
People with diabetes and blood pressure above 140/90 mm Hg are at a higher risk for cardiovascular, kidney, and eye complications.	Preventing, detecting, and treating acute and chronic complications	58.3	0.55	<0.001
Exercising regularly, meditating or relaxing more frequently, and managing a healthier diet to control diabetes can help lower high blood pressure.	Preventing, detecting, and treating acute and chronic complications	52.8	0.553	<0.001
50% of people with diabetes have high blood pressure.	Preventing, detecting, and treating acute and chronic complications	76.2	0.367	<0.001
Negative emotions can induce stress that produces discomfort, which can eventually cause more health problems.	Healthy coping with psychosocial issues and concerns	70.7	0.395	<0.001
The best way to cope with the disease is to have an active approach and work to face the problem head-on and seek a solution.	Healthy coping with psychosocial issues and concerns	76.5	0.461	<0.001
Stress increases blood glucose levels.	Healthy coping with psychosocial issues and concerns	87	0.382	<0.001

Legend: Sample size of 306; items included if point biserial ≥ 0.15 and percentage of respondents answering the question correctly ≥ 50.1%; items outside of these criteria were reviewed and/or eliminated.

**Table 4 healthcare-11-02117-t004:** Kuder-Richardson Formula 20 (KR-20) scores for subscales.

	Number of Items	KR-20
Diabetes disease process and treatment	5	0.46
Healthy eating	9	0.42
Physical activity	5	0.26
Monitoring and using patient generated health data	7	0.09
Medication use	8	0.45
Problem solving	4	0.6
Preventing, detecting, and treating acute and chronic complications	8	0.42
Healthy coping with psychosocial issues and concerns	8	0.34
Behavior scale	24	0.62
Knowledge scale	30	0.64
Entire scale	54	0.78

**Table 5 healthcare-11-02117-t005:** The relationship between diabetes knowledge and sociodemographic variables on diabetes management behavior.

	Coefficients	95% CI		*p*-Value
Knowledge Score	0.46	0.336	0.584	<0.001
Latest HbA1c results	−0.055	−0.383	0.272	0.739
How long [in months] diagnosed with T2DM	−0.005	−0.015	0.005	0.335
Gender	0.457	−0.077	0.992	0.093
Education	0.586	0.143	1.03	0.01
Geographic setting (urban/rural)	0.097	−1.128	1.321	0.876
Constant	4.985	0.844	9.126	0.019

CI: confidence interval.

## Data Availability

Data may be requested from corresponding author.

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
