# Peer review of "The Development of a New Tool to Help Patients and Their Providers Evaluate Self-Management of Type 2 Diabetes Mellitus"

_healthcare, 2023, doi:10.3390/healthcare11152117_

Round 1
Reviewer 1 Report
Comments to the manuscript “The development of a new tool to understand diabetes self-management.”
An important issue is addressed in the paper. Knowledge about diabetes equips individuals with the ability to analyze situations, identify potential problems, and make informed decisions to manage their blood sugar levels effectively. There are some aspects of the paper that require further clarification.
The title of the paper may not accurately reflect its content. It is unclear whether the developed tool would improve diabetes self-management understanding. Understanding implies knowing and comprehending the nature or meaning of. The title may be based on the objective stated at the end of the introduction. - the objective of this study is to develop a diabetes self-management assessment tool that allows patients to self-evaluate T2DM knowledge and self-management behavior-.
In the Method section, the design subsection is written: This study was conducted using a cross-sectional online survey with a patient panel. It is unclear which panel the authors are referring to.
The study was reviewed by the University of Mississippi Institutional Review Board (IRB) before survey administration. Please provide an identification number for the study´s approval .
Sample selection requires further explanation. How was the sample selected? How many people were invited to participate in the survey? Which were the inclusion and exclusion criteria?
It is mentioned that the minimum target sample size for this descriptive pilot study was 50 participants. It is necessary to explain this further including the formula and data used to calculate the sample size. Would this be considered a pilot study?
The survey consisted of eight sections and a total of 56 questions, which were designed to measure participants' current knowledge and behavior regarding diabetes self-management. No information was provided regarding the internal consistency of each of the eight sections of the survey instrument.
A key aspect of the study is the study group selection. An examination of some of the biases that may result from sample selection would be useful.
There were 420 responses, of which 306 were completed questionnaires. Which impact would the loss of approximately 27% of the responses have on the study findings' validity? Additionally, it should be noted that the sample included diabetes and prediabetic participants. How many participants were in each group and was this aspect explored in the responses to the survey instrument?
In the results, it is reported that 55.9% and 22.9% of the participants had a master’s degree or a doctorate degree, respectively. How does this affect the external validity of the findings?
Was there any attempt to associate the HbA1c test results with the different sections of the survey instrument? This is significant because glucose control is a key element in DM2 management.
Table 4 presents the regression coefficients between diabetes knowledge and diabetes management behavior. The results indicated no association with the latest HbA1c results. Why was glucose level not associated with diabetes management behavior? It would be interesting to discuss this finding. Age is a crucial variable in diabetes management. The model does not include this variable in Table 4, but it is recommended that it be included. Additionally, why was glucose level not associated with diabetes management behavior? It would be interesting to discuss this finding.
Finally, in the discussion section, it is indicated that the main objective of this project was to develop a patient decision aid to assess patients´ educational knowledge and behavior toward diabetes self-management. The title, the objective stated at the end of the introduction, and the above information in the discussion section appear to be inconsistent.

There is no problem with the quality of English.
Reviewer 2 Report
Review Comments
In order to discuss The development of a new tool to understand diabetes self- management. The article has a clear idea and a rigorous structure. If the author can improve the following questions, the structure of the article will be more complete:
1. The title of the article should specify that the study subjects were patients with T2DM.
2.. The abstract of this article lacks a background description of diabetes self-management. Please add relevant knowledge background to make the article more complete.
3.In the part of “2.2 Participants”,the inclusion criteria of participants lacked relevant evidence.Furthermore, Please explain why the age range is set at less than 18 years and elaborate on the screening method.
4.The study lacked clinical study registration.
5. In the part of “4.Discussion ”,line244-250, The author points out some defects of this study and I agree.But could these defects be ignored?
6.In the part of Table 1,the author mentioned that“137 participants did not report their latest HbA1c.”Please think about whether this makes a difference to the results.
Author Response
Please see that attachment
